# Prevalence of neurocysticercosis and its characteristics among people with epileptic seizures and progressively worsening severe headaches in 60 villages in three provinces of Burkina Faso

Athanase Millogo[1,2☯], Veronique Dermauw[3☯], Rasmané Ganaba[4], Pierre Dorny[3], Zékiba Tarnagda[5], Rabiou Cissé[6], Marie-Paule Boncoeur-Martel[7], Vivien Richter[8], Andrea S. Winkler[9,10], Hélène Carabin[11,12,13,14]*

1 Centre Hospitalier Universitaire Souro Sanou, Bobo-Dioulasso, Burkina Faso, 2 Université Ouagadougou, Joseph Ki-Zerbo, Ouagadougou, Burkina Faso, 3 Department of Biomedical Sciences, Institute of Tropical Medicine, Antwerp, Belgium, 4 AFRICSanté, Bobo-Dioulasso, Burkina Faso, 5 Institut de Recherche en Sciences de la Santé, Ouagadougou, Burkina Faso, 6 Department of Radiodiagnosis and Medical Imagery, Yalgado Ouedraogo University Hospital Center, Ouagadougou, Burkina Faso, 7 Hôpital Universitaire Dupuytren, Limoges, France, 8 Diagnostic and Interventional Neuroradiology, Department of Radiology, University Hospital Tuebingen, Tübingen, Baden-Württemberg, Germany, 9 Department of Neurology, TUM University Hospital, and Center for Global Health, School of Medicine and Health, Technical University of Munich (TUM), Munich, Germany, 10 Department of Community Medicine and Global Health, Institute of Health and Society, University of Oslo, Oslo, Norway, 11 Department of Pathology and Microbiology, University of Montreal, Montreal, Canada, 12 Centre de Recherche en Santé Publique (CReSP), Montreal, Canada, 13 Groupe de Recherche en Épidémiologie des Zoonoses et Santé Publique (GREZOSP), Montreal, Canada, 14 Department of Social and Preventive Medicine, University of Montreal, Montreal, Canada

☯ These authors contributed equally to this work.
* helene.carabin@umontreal.ca

## Abstract

### Purpose

Neurocysticercosis (NCC) is a common cause of epilepsy in low- and middle-income countries (LMICs). Few studies have described neuroimaging findings in individuals with headaches in addition to epilepsy. Our aim was therefore to describe the types of NCC lesions seen on cerebral computed tomography (cCT), among participants with progressively worsening severe headaches (PWSH) and epilepsy living in 60 villages in Burkina Faso, and to determine the prevalence of NCC in these groups,.

### Methods

Data from a screening questionnaire for epilepsy and PWSH and subsequent neurological examination were gathered as part of a baseline cross-sectional component of a cluster randomized controlled trial conducted between February 2011 and January 2012. Those screening positive and a sub-sample of individuals screening negative were investigated by one of the two physicians in the field. Participants for whom

**Data availability statement:** All data underlying the findings are reported in the manuscript.

**Funding:** Financial support for this work was provided by the National Institute of Neurological Disorders and Stroke (NINDS) and by the Fogarty International Center (FIC) of the National Institutes of Health (NIH) under the Brain in the Developing World: Research across the life span program, Grant R01NS064901 (grant received by HC). The funders had no role in study design, data collection and analysis, decision to publish, or preparation of the manuscript.

**Competing interests:** The authors have declared that no competing interests exist.

the physician confirmed the presence of the neurological signs/symptoms of interest were invited for cCT. Participants receiving cCT were tested for the presence of circulating cysticercal antigens using the B158/B60 Ag-ELISA and of antibodies using the rT24H EITB. For all individuals undergoing cCT, observed lesions were radiology categorized as active, degenerated or inactive based on recommendations from the literature, or uncertain, when radiologists were unsure whether they met the definitions. These individuals were finally clinically categorized as definitive or probable cases of NCC according to the internationally recognized diagnostic criteria for NCC, or as uncertain cases of NCC when radiologists were unsure about the lesions.

## Results

A total of 249 individuals were diagnosed with the neurological signs/symptoms of interest by the physicians. Upon further review by the neurologist, 9 were found to have no neurological signs/symptoms of interest, 109 were diagnosed with epilepsy, 116 with PWSH, and 15 with both. In total, there were 17 definitive NCC cases, 13 probable cases, and 15 cases with an uncertain NCC diagnosis. Among individuals with epilepsy and PWSH, the prevalence of NCC (based on the clinical categorization as definitive or probable NCC cases) was 16.9% (21/124, 95% CI: 10.8%-24.7%) and 7.6% (10/131, 95% CI: 3.7%-13.6%), respectively. Of the definitive or probable NCC cases, 46.7% tested positive for the rT24H EITB (14/30), and 33.3% had a positive Ag-ELISA result (10/30). Among these cases, there were a total of 19 with active lesions, of which 11 showed viable lesions with a scolex, and 6 without, and 2 with degenerating lesions), whereas 29 had inactive (calcified) lesions including 11 mixed lesions.

## Discussion

This study is one of the first to provide an estimate for the prevalence of NCC in people with PWSH on the African continent. Furthermore, the findings confirm significant regional variation in the prevalence of NCC cases in people with epilepsy and underscore the need for further research into the inflammatory response to *T. solium* larvae. Limitations included diagnostic challenges, time delays between assessments, and potential underestimation of the prevalence of NCC due to neuroimaging constraints.

## Author summary

Neurocysticercosis (NCC) is a parasitic infection of the brain caused by the larvae of *Taenia solium*, a zoonotic tapeworm transmitted between humans and pigs. It is a major cause of epilepsy and possibly also associated with severe headaches, particularly in low- and middle-income countries where the parasite is common. This study investigates the prevalence of NCC in people with

epilepsy and those experiencing progressively worsening severe headaches (PWSH) in 60 villages in Burkina Faso. To determine how often NCC occurs in these individuals, we used brain imaging (CT neuroimaging) and blood tests to detect markers of infection. Among 124 people with epilepsy, 16.9% were diagnosed with definitive or probable NCC. In contrast, among 131 individuals with PWSH, 7.6% were found to have NCC. Most NCC cases showed calcified brain lesions, which are thought to trigger seizures and headaches through inflammation.

## Introduction

Neurocysticercosis (NCC) is a neglected parasitic zoonosis of the central nervous system, resulting from infection of the brain or spinal cord with the metacestode larval form of *Taenia solium*. Depending on the localization of the parasite in the brain, NCC may cause a variety of neurological signs and symptoms, epileptic seizures, epilepsy and headaches being the most common clinical manifestations, or may be clinically asymptomatic [1–5].

In endemic areas, NCC is regarded as the great imitator because it can mimic almost any cerebral disorder and adult-onset seizures are highly suggestive of NCC [2]. A meta-analysis published in 2010 estimated that in those areas, 29.0% (95%CI: 22.9%-35.5%) of people with epilepsy had NCC lesions at neuroimaging [6]. In sub-Saharan Africa, several more recent studies have reported a variable, yet often high prevalence of NCC in people with epilepsy [7–13]. Headaches have been reported as the second commonest manifestation of symptomatic NCC in patients seen in neurology clinics, as found in another systematic review [1]. However, we are aware of only one study with low risk of bias reporting the proportion of NCC lesions among people with severe headaches in a community-based setting. That study was conducted in a rural community of Ecuador in 1992 where the proportion of NCC detected by CT was 28% (Bayesian Credible Interval: 18.4%-39%) among people with migraine or tension headaches [14]. In a pilot study conducted in three villages of Burkina Faso, of which one was known to have a high number epilepsy cases, the prevalence of NCC among people with epilepsy was shown to be more than 40% in two villages but absent in the third one [15,16]. This small-scale study did not include participants with headaches and had used a convenient sampling strategy to select villages which could have been biased. Later on, a large-scale cluster randomized controlled trial was set up in 60 villages in three provinces in Burkina Faso, aiming to estimate the effectiveness of an educational intervention to reduce the incidence of cysticercosis in humans [17]. The baseline cross-sectional component of this trial unraveled that the village-level prevalence of human cysticercosis ranged from 0 to 11.5% [18]. Moreover, neurological screening was able to establish positive associations between seropositivity to cysticercal antigens and active epilepsy (odds ratio (OR): 2.4; 95% confidence interval (CI): 1.15-5.00) and progressively worsening severe headaches (PWSH) (OR: 2.59; 95%CI: 1.34-4.99; 21). The present study aimed to describe the types of NCC lesions seen at neuroimaging among individuals with epilepsy and PWSH in these 60 villages in Burkina Faso, and estimate the proportion of NCC in these groups.

## Methods

### Ethics statement

This study was approved by the Institutional Review Board at the University of Oklahoma Health Sciences Center (IRB#1419) and the ethical review panel at the Centre MURAZ (Burkina Faso) (14–0027-AFRICSANTE/DR). All participants were given written consent forms that were read by the field staff. Consent forms were signed or marked with an "X" or a fingerprint by those who were unable to write. All consent forms were signed by a witness. For those less than 16 years of age but 5 or greater, parents gave written consent for their children. Assent was asked from individuals aged 10–15 years. The field staff answered participants' questions. As an incentive to participate in the study, all participants were offered a bar of soap. Additionally, all patients received standard care according to the national guidelines. Because the neuroimaging results were not available at the time of the examination by one of the two physicians, symptomatic

treatment was provided based on the presenting neurological signs and symptoms. Participants with PWSH were provided with paracetamol for a few weeks, whereas participants with epileptic seizures received phenobarbital for at least 6 months. During this period, the field team was around and in close contact with the involved nurses in the study villages, who monitored any side effects of the medication and contacted the medical team for assistance or advice as needed.

## Study design

This study is part of the baseline cross-sectional component of a large-scale cluster randomized controlled trial that aimed to reduce the incidence of human cysticercosis through the implementation of an educational program [17].

## Setting and participant selection

For the present study, the study sample entails those participants of the baseline cross-sectional component of the trial with confirmed neurological signs or symptoms of interest (epilepsy and PWSH) who underwent cerebral computed tomography (cCT), allowing us to describe the types of NCC lesions seen at neuroimaging among individuals within these 60 villages in Burkina Faso, and estimate the proportion of NCC in these groups. Participant selection for the present study was based on a two-step process: a screening questionnaire followed by diagnostic confirmation by the study physicians.

The baseline component of the trial took place between February 2011 and January 2012. The study area and selection of study villages and concessions (i.e., grouped households) are described in detail in Carabin et al. [18]. Briefly, in three provinces in Burkina Faso (Boulkiemdé, Nayala and Sanguié), a total of 60 eligible villages (eligibility criteria: being on the census map, with at least 1000 inhabitants, where pigs are raised, not on a major road, separated from another village by at least 5 km) were randomly selected in all 30 pig-raising departments. In each village, 80 concessions were selected using a stratified random sampling approach. Participant selection for the larger study was previously described [19,20]. In brief, one individual per concession was invited to participate in the study, resulting in 80 people being sampled in 80 concessions. Each individual was invited to answer a questionnaire including screening questions for the presence of epileptic seizures, epilepsy and PWSH (for questionnaire, see Sahlu et al. [21]). Age, gender, educational level and wealth quintile were also recorded as described in Carabin et al. [18]. Participants were asked to consent to provide a blood sample until 60 individuals agreed in each village.

Individuals answering "yes" to at least one screening question for seizure/epilepsy or PWSH were examined by one of the two study physicians for confirmation, and confirmed cases were invited to provide a blood sample if they had not already provided one. For the purpose of the cluster randomized control trial, another individual was invited to participate from the household where the original participant was confirmed to have the neurological signs/symptoms of interest. In addition, a sub-sample of individuals screening negative to the questionnaire were also examined by the physicians (see Sahlu et al. [21] for more details). All cases confirmed by the physicians to suffer from the neurological signs/symptoms of interest (336 cases) were invited to undergo cCT in the capital city Ouagadougou. To avoid stigmatization, confirmed cases from several villages were invited to meet in another location to take a bus to Ouagadougou. The fear of stigma also resulted in conducting two rounds of visits by the two physicians to be able to examine all of those who had screened positive. Most of the second round took place between March and May 2013. The CT machine was broken for some time during the course of the study. This resulted in delays between the time of the physicians' examination and the cCT. For the present study, the final study sample entails those participants with confirmed neurological signs/symptoms of interest who underwent cCT.

## Measurement and definitions of epilepsy and progressively worsening severe headaches

The presence of epilepsy and PWSH were determined by the physicians based on a clinical questionnaire. Epilepsy was defined as having had at least two apparently unprovoked seizures of central nervous system origin, occurring more than 24 hours apart, and confirmed by the study physicians through history and neurological evaluation according to the

International League Against Epilepsy classification [22]. PWSH were defined as symptoms being progressively worsening in severity, currently occurring more than once a week for more than two weeks, with each episode lasting for at least three hours and with a pain intensity affecting the person's ability to work, play, attend school or carry out usual activities or that requires analgesics. This definition was an adapted version of the headache-definition of Jensen and Stovner [23], allowing to capture NCC related headaches [24]. In case of doubt, the physicians called the neurologist to discuss the case, until consensus was reached.

Several months after the examination by the physicians, the original diagnosis was revised by the study neurologist, upon reviewing the electronic database and comments left on the Personal Digital Assistant (PDA, i.e., a handheld device for information storage), by the physicians.

### Blood analyses

Blood samples of participants were analyzed for the presence of antigens of the larval stage of *T. solium*, as an indicator for infection with viable cysticerci. Antigens were measured using the B158/B60 enzyme-linked immunosorbent assay (Ag-ELISA) [25]. Blood samples were also examined for the presence of antibodies against cysticerci, thus indicating present and past exposure or infection with cysticerci that may not be viable anymore. Antibodies were measured using the lentil lectin-bound glyco-protein enzyme-linked immunoelectrotransfer blot assay (LLGP-EITB) using the recombinant T24H (rT24H) Ag [26]. Considering its perfect agreement with the conventional EITB, this test was confirmed to serve as a valuable alternative to the native antigen LLGP-EITB for the diagnosis of NCC in the same large-scale trial [20].

### Cerebral computed tomography and radiological classification of neurocysticercosis

All participants confirmed to suffer from epilepsy and PWSH based on the physicians' evaluation, were invited to undergo cCT, performed using a General Electric 8 Barrett LightSpeed scanner (General Electric) at the Centre Médical Schiphra in Ouagadougou. We performed a biphasic helical acquisition, both without and with injection of contrast medium, using native slices of 5 mm thickness, multiplanar reconstructions, and image readings in parenchymal and bone windows. Telebrix 35 (50 mL; Guerbet, Aulnay-sous-Bois, France), the only contrast medium available in Burkina Faso at the time, was used as iodinated contrast medium. It was administered intravenously, with image acquisition 5–10 min later to allow for adequate cerebral enhancement. Prior to the injection, the renal function was verified by measuring serum creatinine levels and calculating the glomerular filtration rate (GFR) using the Cockcroft–Gault formula [27]. All patients had a GFR greater than 30 mL/min/1.73 m², allowing the injection of contrast medium without risk of contrast-induced nephropathy. No participant experienced an allergic reaction to the iodinated contrast medium. At the time of our study, a magnetic resonance imaging (MRI) facility was not available in the country. The study radiologist used a standardized form to report the type, number and location of NCC lesions in addition to other abnormalities. Two other radiologists independently reviewed all cCT scans for the presence, number and location of lesions by type. Observed lesions were radiologically categorized as lesions highly suggestive of NCC, characterized as either active (viable cysts: vesicular stage with or without scolex; degenerating cysts: colloidal or granular nodular stage) or inactive (calcified stage) [28,29]; or uncertain NCC lesions, in case the radiologists were unsure if they met the descriptions included in the neuroimaging criteria for NCC diagnosis [4].

Any disagreements were resolved by email and when a consensus could not be reached, the cCT scans were further evaluated by a neurologist with extensive experience in NCC.

### Definition of clinical neurocysticercosis

The clinical categorization of cases as definitive or probable NCC cases was based upon the revised diagnostic criteria for NCC [4]. The only absolute criterion we evaluated in our participants was evidence of cystic lesions showing a scolex on the cCT scan. We considered cystic lesions without discernible scolex, single or multiple rings or nodular enhancing

lesions and small parenchymal round calcifications on neuroimaging studies as major neuroimaging criteria. As cCT was performed only once, we could not evaluate resolution of lesions spontaneously or after albendazole (i.e., confirmative neuroimaging criteria). Minor neuroimaging criteria were obstructive hydrocephalus (symmetric or asymmetric) or abnormal enhancement of basal leptomeninges. Furthermore, as all cases had clinical manifestations suggestive of NCC (i.e., seizure/epilepsy and/or PWSH), and were living in a *T. solium* endemic area, they all fitted one minor clinical and one minor exposure criterion, respectively. A positive test result for one or both of the blood tests, thus detection of specific anticysticercal antibodies or cysticercal antigens, was considered a major clinical/exposure criterion.

This categorization as definitive or probable NCC case was only done for those participants with lesions highly suggestive for NCC as determined by the radiologists. Individuals with uncertain NCC lesions, i.e., lesions for which the radiologists were unsure whether they met the descriptions of the neuroimaging criteria for NCC diagnosis [4], were categorized as uncertain NCC cases. To reflect the diagnostic uncertainty, and because we consider them neither confirmable as NCC case nor were they confidently ruled out, these cases were described separately from the definitive and probable cases.

### Data management and statistical analysis

The physicians entered their examination results onto a PDA. These were transferred onto an Excel spreadsheet for verification by the neurologist several months later. The radiological readings were initially entered on a paper form and then captured on an Excel spreadsheet with predetermined fields. This formatted spreadsheet was used by the other two radiologists to enter their readings.

Descriptive statistical analyses were run. This entailed the calculation of frequencies, proportions and associated 95% exact confidence intervals. All statistical analyses were conducted using the R software [30].

## Results

### Neurological screening

A total of 4,970 people were screened for epileptic seizures and PWSH, among whom 880 individuals were examined by one of the two physicians in the field. Of those, 336 were confirmed to suffer from the neurological signs/symptoms of interest, and 249 individuals (74%) underwent cCT examination, 116 with epilepsy, 112 with PWSH and 21 with both. Upon file revision by the neurologist, nine individuals were determined to not be confirmed with the neurological signs/symptoms of interest, while 109 were deemed to suffer from epilepsy, 116 from PWSH and 15 from both.

### Clinical categorization

Overall, based on the clinical categorization, there were 17 definitive, and 13 probable NCC cases. There were another 15 cases for which the NCC diagnosis was uncertain, whilst the remaining 204 individuals were categorized as non-NCC cases.

### Socio-demographic characteristics of and serological results in NCC vs. non-NCC cases

Most definitive and probable NCC cases were male (21/30, 70.0%) (Table 1), whereas the non-NCC cases were mostly female (138/204, 67.7%). The median age of definitive and probable NCC cases was 50 years (interquartile range (IQR): 61 years) versus 34 years (IQR: 64) in the individuals categorized as non-NCC case. Most definitive and probable NCC cases were from the lower three wealth quintiles (22/30, 73.3% versus 130/204, 63.7% in non-NCC cases), and did not finish primary school (23/30, 76.7% versus 156/203, 76.9% in non-NCC cases).

Among the definitive and probable NCC cases, 46.7% had a positive result for the rT24H EITB (14/30), whereas only 1.5% of the non-NCC cases had the same test result (3/196). A positive Ag-ELISA result was found in 33.3% of definitive and probable NCC cases (10/30), but in only 3.5% of non-NCC cases (7/203).

**Table 1. People with epilepsy, progressively worsening severe headaches and their clinical categorization as definitive/probable cases of neurocysticercosis, identified in 60 villages in Burkina Faso, 2011-2012.**

| Age | Gen-der | Neurological signs/ symptoms | | Type of lesions | | | | | Serology | | Degree of certainty | Inter-val (days) |
| | | Evalua-tion phy-sicians | Evaluation neurolo-gist | N active (via-ble, vesicular with scolex) | N active (viable, vesicular with-out scolex) | N active (degen-erating) | N inactive (calcified) | Hydro-cepha-lus | Ag-ELISA | rT24H | | |
|---|---|---|---|---|---|---|---|---|---|---|---|---|
| 47 | M | E | E | 27 | 11 | 0 | 24 | – | + | + | Def | 21 |
| 44 | M | E | E | 0 | 0 | 0 | 2 | – | + | + | Def | 21 |
| 52 | M | E+PWSH | PWSH | 1 | 0 | 0 | 5 | – | – | – | Def | 24 |
| 61 | M | E | E | 10 | 0 | 1 | 16 | + | + | + | Def | 25 |
| 35 | F | E+PWSH | E+PWSH | 0 | 0 | 0 | 15 | – | – | + | Def | 44 |
| 12 | M | E | E | 0 | 0 | 0 | 2 | – | – | + | Def | 14 |
| 47 | F | E | E | 10 | 4 | 0 | 15 | – | – | + | Def | 63 |
| 60 | F | PWSH | PWSH | 0 | 0 | 0 | 2 | – | + | + | Def | 56 |
| 12 | M | E | none | 0 | 7 | 0 | 3 | + | – | – | Def | 63 |
| 67 | M | E | E | 1 | 0 | 0 | 13 | + | – | + | Def | 24 |
| 34 | M | E | E | 29 | 19 | 0 | 53 | – | + | + | Def | 23 |
| 51 | F | E | E | 0 | 0 | 0 | 10 | – | – | + | Def | 31 |
| 65 | M | E+PWSH | E+PWSH | 1 | 0 | 0 | 0 | – | + | + | Def | 44 |
| 59 | M | E | E | 2 | 0 | 0 | 8 | – | – | + | Def | 18 |
| NA | M | E | E | 4 | 0 | 0 | 6 | – | + | – | Def | 116 |
| 50 | M | E | E | 10 | 2 | 0 | 4 | – | + | + | Def | 117 |
| 61 | F | E+PWSH | E+PWSH | 1 | 1 | 1 | 3 | – | + | + | Def | 98 |
| 35 | M | E | E | 0 | 0 | 0 | 1 | – | – | – | Prob | 44 |
| 51 | M | E | E | 0 | 0 | 0 | 1 | – | – | – | Prob | 40 |
| 58 | M | E | E | 0 | 0 | 0 | 6 | – | – | – | Prob | 47 |
| 6 | F | E | E | 0 | 0 | 0 | 1 | – | – | – | Prob | 173 |
| 44 | M | E | E | 0 | 0 | 0 | 1 | – | – | – | Prob | 9 |
| 29 | F | PWSH | PWSH | 0 | 0 | 0 | 2 | – | – | – | Prob | 17 |
| 49 | M | E | none | 0 | 0 | 0 | 1 | – | + | – | Prob | 120 |
| 56 | M | PWSH | PWSH | 0 | 0 | 0 | 4 | – | – | – | Prob | 116 |
| 54 | M | E | E | 0 | 0 | 0 | 6 | – | – | – | Prob | 124 |
| 63 | F | PWSH | PWSH | 0 | 0 | 0 | 3 | – | – | – | Prob | 155 |
| 50 | M | PWSH | PWSH | 0 | 0 | 0 | 1 | – | – | – | Prob | 99 |
| 52 | F | PWSH | PWSH | 0 | 0 | 0 | 3 | – | – | – | Prob | 99 |
| 41 | M | E | E | 0 | 0 | 0 | 5 | – | – | – | Prob | 120 |

E: epilepsy, G: gender (M: male, F: female), PWSH: progressively worsening severe chronic headaches, Def: definitive, Prob: Probable

Interval: interval between screening questionnaire and cCT in days

The median interval between screening questionnaire and cCT was 31 days in the definitive NCC-group (IQR: 40 days), whereas in the probable NCC group, the median interval was 99 days (IQR: 76 days). In the non-NCC case group, the median interval was 57 days (IQR: 94 days).

## Radiological findings in the definitive, probable and uncertain NCC cases

Amongst the 30 definitive and probable NCC cases, active lesions were detected in 19. Of these, a total of 11 cases had viable vesicular lesions with scolex, six cases had viable vesicular lesions without a visible scolex, and another two had

degenerating lesions. All but one definitive and probable NCC cases had at least one calcified lesion. Combinations of the different lesions types were found in 11 of the definitive and probable NCC cases. Single cysts were found in seven definitive and probable NCC cases. One of the definitive and probable NCC cases had 101 lesions in the brain, of which 29 cysts with scolex, 19 cysts without scolex, and 53 calcifications. For this case, the NCC lesions were located in the frontal (n = 33), temporal (n = 27), parietal (n = 19) and occipital lobes (n = 17). Other locations of lesions reported for the definitive and probable NCC case group were the centrum ovale, cerebellum, peduncle, intraventricular and basal ganglia. Hydrocephalus was observed in three definitive and probable NCC cases, none showed enhancement of leptomeninges. Among the 15 uncertain NCC cases, nine had a single lesion, the others had two or three lesions (Table 2). All these lesions had a calcified appearance. None of the uncertain NCC cases had hydrocephalus or enhancement of leptomeninges.

### Prevalence of the neurological signs/symptoms of interest among NCC cases and link with radiological findings

Based on the final classification of the neurological signs/symptoms of interest by the neurologist, the prevalence of a clinical categorization as definitive or probable NCC case among people with epilepsy and PWSH was 16.9% (21/124, 95%CI: 10.8%-24.7%) and 7.6% (10/131, 95%CI: 3.7%-13.6%), respectively. For clarity, individuals who presented with both epilepsy and PWSH were included in both the epilepsy and PWSH categories, as the objective was to estimate the prevalence of NCC within each clinical manifestation rather than in mutually exclusive groups. Consequently, the denominators for these prevalence estimates were 124 for epilepsy and 131 for PWSH, yielding a total of 249 participants with valid cCT results. Of note, only one patient with PWSH was definitely diagnosed with NCC. When uncertain NCC cases were included, the prevalence of NCC among people with epilepsy, and PWSH increased to 22.6% (28/124, 95%CI: 15.6%-31.0%) and 14.5% (19/131, 95%CI: 9.0%-21.7%), respectively. Twenty out of the 21 people with epilepsy, and 9 out of 10 with PWSH, who were classified as definitive or probable NCC case, had calcified lesions (95.2%, 95%CI: 76.2%-99.9%; 90.0%, 95%CI: 55.5-99.7).

**Table 2. People with epilepsy, progressively worsening severe headaches and their clinical categorization as uncertain cases of neurocysticercosis, identified in 60 villages in Burkina Faso, 2011-2012.**

| | | Neurological signs/symptoms | | | Serology | | |
|---|---|---|---|---|---|---|---|
| Age | Gender | Evaluation physicians | Evaluation neurologist | Number of lesions | Ag-ELISA | rT24H | Interval (days) |
| 39 | M | E | E | 1 | – | – | 24 |
| 54 | M | E+PWSH | E+PWSH | 3 | – | – | 25 |
| 25 | M | E | E | 1 | – | – | 40 |
| 48 | M | E | E | 1 | – | – | 40 |
| 60 | F | PWSH | PWSH | 1 | – | – | 149 |
| 57 | M | E+PWSH | E+PWSH | 2 | – | + | 16 |
| 35 | M | PWSH | PWSH | 1 | – | – | 17 |
| 51 | F | PWSH | PWSH | 1 | – | – | 16 |
| 44 | F | PWSH | PWSH | 3 | – | – | 16 |
| 45 | F | E | E | 1 | – | – | 16 |
| 60 | M | E | E | 1 | – | + | NA |
| 8 | F | PWSH | none | 1 | – | + | 124 |
| 52 | F | PWSH | PWSH | 2 | – | – | 123 |
| 59 | M | PWSH | PWSH | 2 | – | – | 119 |
| 64 | M | PWSH | PWSH | 2 | – | – | 81 |

E: epilepsy, G: gender (M: male, F: female), PWSH: progressively worsening severe headaches

Interval: interval between screening questionnaire and cCT in days

## Discussion

This study aimed to describe the type of NCC lesions seen on neuroimaging in people with epilepsy and PWSH in 60 villages in Burkina Faso, and to estimate the prevalence of NCC in these groups. Few community-based studies of this scale have investigated the contribution of NCC to epilepsy, and even less that to PWSH in endemic areas making this estimation clinically and programmatically relevant. This is especially important since epilepsy and headaches are the most common neurological manifestations of symptomatic NCC [1] and result in important monetary and societal burden for the patients and their family [31–33]. Unravelling the frequency of NCC in these subgroups will therefore assist in informing differential diagnostic decision-making, in prioritizing resource allocation (e.g., neuroimaging and treatment), and in designing targeted interventions in endemic settings. Although estimating the prevalence of NCC prevalence in the general population would be of interest, it is unethical to conduct cCT among individuals who do not present with neurological signs/symptoms. In addition, such information would have limited immediate clinical utility and not directly benefit participants.

For people with epilepsy, a systematic review conducted for the period 1990–2008 estimated the prevalence of NCC at 31.7% (95% CI: 25.6%-38.2%) for community-based studies, based on eight records, with study-specific estimates ranging between 18.0% and 54.0% [6]. In that estimate, only one study performed on the African continent was included (i.e. 25.0% for a pilot study conducted in three villages in Burkina Faso) [16]. More recent community-based estimates for the prevalence of NCC in people with epilepsy from the continent ranged between 4.0% and 54.2% [8–12,34]. In our study, it was estimated at 16.9% (95%CI: 10.8%-24.7%) when only definitive or probable NCC cases were considered but increased to nearly 23% when the uncertain NCC cases were included. Up to now, even fewer community-based studies have looked into the prevalence of NCC among people with headaches. To the best of our knowledge, only two earlier small-scale community-based studies have been conducted, one in Ecuador, estimating this proportion at 33.3% (19/57) [14], and a more recent one in Zambia, estimating the proportion at a very similar 33.0% (15/45) [35]. In our study, this prevalence was estimated at a lower 7.6% (10/131, 95%CI: 3.7%-13.6%), yet our study only looked at one type of headaches (PWSH).

All but one person with epilepsy categorized as definitive or probable NCC case had at least one calcified lesion. Only one of the earlier studies performed in sub-Saharan Africa provided a similar estimate for the prevalence of calcified NCC (i.e., the presence of calcified lesions in NCC cases), in people with epilepsy. In a community-based study in Zambia, all 7 people with epilepsy classified as NCC case (100%), had calcified lesions. In another study outside the continent, in Bolivia [36], 19 out of 31 people with epilepsy (61.3%) who were classified as definitive/probable NCC cases, had calcified lesions. Up to now, it is assumed that epileptic seizures in NCC cases are mainly triggered by the inflammatory response to dead or dying *T. solium* larvae, and this would occur more commonly when degenerating or calcified cysts are present [37]. Nevertheless, epilepsy has also been observed in NCC cases with active lesions (i.e., with vesicular cysts) and calcified NCC without perilesional inflammation. However, perilesional edema caused by inflammation might be hard to detect on neuroimaging due to temporary spikes in the host's inflammatory response, or perhaps due to the larval cysts releasing molecules that trigger seizures without causing inflammation [37]. Overall, this points to presence of knowledge gaps on the development of epilepsy related to NCC, especially in case of active lesions or calcified lesions without perilesional inflammation [37,38].

In our study, all but one PWSH had calcified NCC. This contrasts with a study conducted in Ecuador where the prevalence of calcified NCC in participants with headaches was estimated at 26.3% (15/57) [14]. However, a more recent community-based survey showed larger frequency of lifetime headaches prevalence (4.18, 95% CI: 1.79–9.75) among calcified NCC cases compared to controls without headaches [39]. Of note, our definition of headaches was not the same as in the two references mentioned. The mechanism by which calcified NCC causes headaches is poorly understood, but it is thought that the perilesional inflammation induces blood–brain barrier disruption, and release of free radicals thus inducing the liberation of the calcitonin gene–related peptide, triggering severe headaches and migraine [39,40].

Antibody detection tests like EITB are valuable for identifying past and recent exposure to *T. solium*, but they have limitations in diagnosing NCC. While seropositivity indicates contact with the parasite, it does not confirm an active infection or brain involvement, as most seropositive individuals show no neurological signs/symptoms. Even in neurologically symptomatic cases, there is a frequent mismatch between test results and neuroimaging findings, with many NCC cases identified on cCT being seronegative [41]. In our study, antibodies were detected in 61.9% of people with epilepsy categorized as definitive or probable NCC case (13/21), similar to the results of a study conducted in Bolivia (65.6%, 21/32) [36]. In our study, antibodies were detected in four out of 10 PWSH categorized as definitive or probable NCC cases (40%). These findings highlight the complexity of using antibody tests alone for diagnosing NCC, especially in regions where transient antibodies or resolved infections are common [41].

This study has several limitations. By design, we focused on the occurrence of NCC in participants with the neurological signs/symptoms of interest. Hence these results are not meant to be generalizable to the general population. Due to our sampling strategy focusing on including participants from 60 villages in three provinces believed to be endemic for cysticercosis, our findings are not representative of the entire country, but rather reflect the patterns in high-risk, pig-raising villages with poor sanitation that met the inclusion criteria of our sampling frame. Furthermore, the diagnosis of potential NCC-related neurological signs/symptoms, such as epilepsy and headaches is not straightforward, especially in a community-based setting. First, and as shown by our team [21], the use of a screening questionnaire for the neurological signs/symptoms of interest has its limitations, and can lead to missing cases suffering from these signs/symptoms. Indeed, the screening questionnaire was estimated to have a sensitivity and specificity of 66.1% and 88.9% for epilepsy and 59.6% and 88.6% for PWSH, respectively [21]. Second, we assumed that the evaluation by the study physicians/neurologist was a reference standard for establishing the presence of the signs/symptoms of interest, which is not the case, especially in a community-based setting without access to a video electroencephalogram (EEG) [42] or standard definitions for severe headaches resulting from NCC. Third, when the study was conducted and up to now, a standard definition and associated standardized questionnaire for NCC-associated headaches have been lacking [13]. Hence, the assessment method for PWSH had its limitations as the definition used in the study was more geared towards chronic headaches as defined in the context of primary headaches. Fourth, the approach for the final neurological diagnosis was based on the evaluation of the neurologist reviewing the original notes of the physicians who had examined the patients. This means that the neurologist also did not examine the patients himself, which has its drawbacks and can compound classification error. However, in this large community-based randomized controlled trial, the estimation of the prevalence of NCC among people with epilepsy and PWSH was a secondary objective, and hence, sending all people diagnosed by the study physicians for routine video EEG, or additional examination by the neurologist, was not feasible with the available funds and with the limited resources in public hospitals of Burkina Faso in 2011. Moreover, we observed a perfect agreement in the diagnostic evaluation of the physicians for 35 individuals who were examined during the two rounds of visits for the medical examinations [21]. For some cases, there was also quite a considerable time interval between the screening questionnaire and cCT due to several reasons, including important population movements [17,43] and possibly to the stigma linked to epilepsy as mentioned above [44]. Hence, for some participants, the neuroimaging results might have changed as compared to those that would have been observed around the time of administration of the screening questionnaire, and of the clinical and neurological evaluation by the physicians, including the possible disappearance of some lesions. This delay may have contributed to the underestimation of viable or degenerating cysts and an overrepresentation of calcified lesions, thereby affecting the clinical classification of NCC. Some cysts could have also completely resolved, resulting in an overall underestimation of the prevalence of NCC in the study population. Such temporal gaps are a common challenge in community-based neuroimaging studies conducted in resource-limited and stigma-prone environments, and should be taken into account when interpreting prevalence and lesion type estimates. In addition, cCT comes with its drawbacks, as it can miss small calcified as well as active lesions [45,46]. Moreover, the classification of some lesions remains subjective, as demonstrated by the large number of rounds of review that took place before an

agreement among radiologists was obtained for some images. This suggests that some level of misclassification error likely remains, especially for single or few calcified lesions.

## Acknowledgments

Our gratitude goes to the EFECAB field team for their hard work in collecting the survey data and to the participants for taking part in the study.

## Author contributions

**Conceptualization:** Athanase Millogo, Rasmané Ganaba, Pierre Dorny, Hélène Carabin.

**Data curation:** Athanase Millogo, Veronique Dermauw, Rasmané Ganaba, Pierre Dorny, Hélène Carabin.

**Formal analysis:** Veronique Dermauw, Hélène Carabin.

**Funding acquisition:** Athanase Millogo, Rasmané Ganaba, Pierre Dorny, Hélène Carabin.

**Investigation:** Athanase Millogo, Rasmané Ganaba, Pierre Dorny, Zékiba Tarnagda, Rabiou Cissé, Marie-Paule Boncoeur-Martel, Vivien Richter, Andrea S. Winkler, Hélène Carabin.

**Methodology:** Hélène Carabin.

**Project administration:** Rasmané Ganaba, Pierre Dorny, Hélène Carabin.

**Resources:** Athanase Millogo, Rasmané Ganaba, Pierre Dorny, Zékiba Tarnagda.

**Software:** Veronique Dermauw, Hélène Carabin.

**Supervision:** Athanase Millogo.

**Writing – original draft:** Athanase Millogo, Veronique Dermauw, Hélène Carabin.

**Writing – review & editing:** Athanase Millogo, Veronique Dermauw, Rasmané Ganaba, Pierre Dorny, Zékiba Tarnagda, Rabiou Cissé, Marie-Paule Boncoeur-Martel, Vivien Richter, Andrea S. Winkler, Hélène Carabin.

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
