## [Decision Letter · Decision Letter 0]

21 May 2025

Prevalence of neurocysticercosis and its characteristics among people with epileptic seizures and progressively worsening severe headaches in 60 villages in three provinces of Burkina Faso

Dear Dr. Carabin,

Thank you for submitting your manuscript to PLOS Neglected Tropical Diseases. After careful consideration, we feel that it has merit but does not fully meet PLOS Neglected Tropical Diseases's publication criteria as it currently stands. Therefore, we invite you to submit a revised version of the manuscript that addresses the points raised during the review process.

Please submit your revised manuscript within 60 days Jul 20 2025 11:59PM. If you will need more time than this to complete your revisions, please reply to this message or contact the journal office at plosntds@plos.org. Please include the following items when submitting your revised manuscript:

We look forward to receiving your revised manuscript.

Kind regards,

Eva Clark, M.D., Ph.D.

Section Editor

Eva Clark

Section Editor

Shaden Kamhawi

co-Editor-in-Chief

Paul Brindley

co-Editor-in-Chief

**Journal Requirements:**

At this stage, the following Authors/Authors require contributions: Athanase Millogo, Rasmané Ganaba, Pierre Dorny, Zékiba Tarnagda, Vivien Richter, Andrea S. Winkler, Veronique Dermauw, and Hélène Carabin. Please ensure that the full contributions of each author are acknowledged in the "Add/Edit/Remove Authors" section of our submission form.

2) Please amend your detailed Financial Disclosure statement. This is published with the article. It must therefore be completed in full sentences and contain the exact wording you wish to be published. Please ensure that the funders and grant numbers match between the Financial Disclosure field and the Funding Information tab in your submission form. Note that the funders must be provided in the same order in both places as well.

**Reviewers' Comments:**

Reviewer's Responses to Questions

**Key Review Criteria Required for Acceptance?**

**Methods**

-Are the objectives of the study clearly articulated with a clear testable hypothesis stated?

-Is the study design appropriate to address the stated objectives?

-Is the population clearly described and appropriate for the hypothesis being tested?

-Is the sample size sufficient to ensure adequate power to address the hypothesis being tested?

-Were correct statistical analysis used to support conclusions?

-Are there concerns about ethical or regulatory requirements being met?

Reviewer #1: Are the objectives of the study clearly articulated with a clear testable hypothesis stated?

Yes

-Is the study design appropriate to address the stated objectives?

Yes

-Is the population clearly described and appropriate for the hypothesis being tested?

Yes

-Is the sample size sufficient to ensure adequate power to address the hypothesis being tested?

looks appropriate

-Were correct statistical analysis used to support conclusions?

yes

-Are there concerns about ethical or regulatory requirements being met?

No

Reviewer #2: Please see below.

Reviewer #3: No- See final comment

**Results**

-Does the analysis presented match the analysis plan?

-Are the results clearly and completely presented?

-Are the figures (Tables, Images) of sufficient quality for clarity?

Reviewer #1: Does the analysis presented match the analysis plan?

yes

-Are the results clearly and completely presented?

yes

-Are the figures (Tables, Images) of sufficient quality for clarity?

yes

Reviewer #2: Please see below.

Reviewer #3: No- See final comment

**Conclusions**

-Are the conclusions supported by the data presented?

-Are the limitations of analysis clearly described?

-Do the authors discuss how these data can be helpful to advance our understanding of the topic under study?

-Is public health relevance addressed?

Reviewer #1: Are the conclusions supported by the data presented?

yes

-Are the limitations of analysis clearly described?

yes

-Do the authors discuss how these data can be helpful to advance our understanding of the topic under study?

yes

-Is public health relevance addressed?

yes

Reviewer #2: Please see below.

Reviewer #3: There is no conclusion.

**Editorial and Data Presentation Modifications?**

Reviewer #1: Accept

Reviewer #2: Please see below.

Reviewer #3: See final comment.

**Summary and General Comments**

Reviewer #1: good study

nicely analyzed

Reviewer #2: Prevalence of neurocysticercosis and its characteristics among people with epileptic seizures and progressively worsening severe headaches (PWSH) in 60 villages in three provinces of Burkina Faso

The manuscript reports part of the findings of a study on NCC in 60 villages in three provinces of Burkina Faso. It estimated NCC prevalence in progressively worsening severe headaches (PWSH) in the rural regions of Burkina Faso. Human neurocysticercosis (NCC) in sub-Saharan Africa (SSA) is recognized as a significant but under-diagnosed and under-reported public health issue. Most of our knowledge comes from specific research projects or hospital settings, which may not represent the general population. The study addresses gaps in NCC epidemiology in sub-Saharan Africa, particularly Burkina Faso, where data from the general population are limited. However, several issues should be addressed by the authors before considering the manuscript for publication.

- This work is the latest of the several publications derived from a project conducted in 2011-2012. The authors are expected to elaborate more about the aspects of NCC in Burkina Faso that are new in this report, and what data were NOT covered in the past six publications.

- According to the manuscript, the aim of the study was to describe the types of NCC lesions seen on cerebral computed tomography (cCT), however, only symptomatic individuals (epilepsy/PWSH) were studied, and the study excluded asymptomatic NCC cases. in addition final diagnosis was based on physician notes review, not direct patient examination, and this risks misclassification. This presents a selection and/or classification bias. Please justify exclusion of asymptomatic individuals and address potential selection bias. Also, the distinction and potential misclassification of “uncertain” NCC cases is underdeveloped.

- please provide more details of CT imaging protocols (slice thickness, contrast use) and radiologist consensus process.

- Delayed CT participation due to epilepsy stigma may bias prevalence estimates. Report attrition rates, e.g., how many refused CT scans due to stigma?

- Findings from 3 provinces out of 45 provinces in the country may not reflect Burkina Faso’s broader epidemiology.

- Only descriptive statistics are used, despite the availability of rich demographic and clinical data. Please consider conducting logistic regression to more in-depth information.

- Median 31–99 days between screening and CT may have altered lesion visibility (e.g., resolved cysts). The limitations should be further emphasized in the context of community-based neuroimaging studies.

- Please justify the rationale for including PWSH. This is only briefly mentioned in the Introduction.

- The link between lesion type (e.g., calcified vs. active) and clinical symptoms is discussed but NOT statistically explored. Please elaborate more on calcified lesion dominance vs. active cysts in relation to seizure/headache mechanisms.

- Please replace the Discussion with "conclusion".

- The citation for Jensen & Stovner (line 138) is not correct (26 > 25).

%

Reviewer #3: The aim of this study was to assess the relationship between symptoms possibly related to NCC (epilepsy and headache) and the diagnosis of NCC. While this may be interesting, there are serious problems with the methodology, the analysis of the results is not complete, and the article lacks relevant conclusions.

In this door-to-door study, a questionnaire relating to neurological symptoms (epilepsy, PWSH) was applied to residents of rural communities. 109 people were diagnosed with epilepsy, 116 with PWSH and 15 with both symptoms. A CT scan with/without contrast was carried out in all these people (240), and in 30 of them a definite (17) or probable (13) diagnosis of NCC was made.

The main problem is the definition of PWSH (lines 133-136). This definition could possibly be compatible with headache in the context of ICH, but not in the context of other NCC presentations. I really don't know on what basis the authors consider this type of headache to be a primary symptom of NCC. It isn't. I'm really very surprised that so many individuals present with this type of symptomatology. Since the individuals included have not been evaluated by a neurologist, serious doubts about the diagnosis can be raised. Headaches are the most common neurological symptoms, and many causes exist, particularly tension headaches. The authors do not specify whether the individuals who suffered from this symptom attended the health center, but if they really did suffer from this type of headache, it is very likely that they received medical attention. No information on this point is available.

The main results of this study are as follows:

- The prevalence of NCC (def or probable) in symptomatic individuals is 12.5% (30/240), 7.1% if only definitive diagnosis of NCC is considered.

- The prevalence of NCC (def or probable) in subjects suffering only from headaches is 5.2% (6/116), with only one individual having a definitive diagnosis of NCC (prevalence: 0.8%).

- The prevalence of NCC (def or probable) in epileptic subjects is 18.3% (20/109), 12% (12/109) if only definitive diagnosis of NCC is considered.

- The prevalence of NCC (def or probable) in subjects with headache + epilepsy is 26.7% (4/15), all with a definitive diagnosis of NCC.

Unfortunately, the authors do not clearly express these results, and the article lacks clear conclusions.

Because of all these shortcomings, I really don't think the interest of these results is sufficient for a research article. Perhaps a short report.

Other comments:

Lines 50-52. Regional variation? This point and its possible causes are not thoroughly discussed in the article.

Lines 77-78: False. In the systematic review mentioned by the authors (the same corresponding author of this work), headaches (and not PWSH) were indeed the second most frequent symptom. But we are talking about two completely different entities. Moreover, as the authors mention in ref. 3, “These estimates are only applicable to patients who are ill enough to seek care in neurology clinics and likely overestimate the frequency of manifestations among all NCC cases”. No information on attention in a health center of the included individuals is available in this article.

Lines 78-80. Headaches are one of the most common neurological symptoms and are not really relevant to the diagnosis of NCC, as evidenced by your results.

Lines 157-158. Contrast injection can cause kidney damage. The authors did not check renal function beforehand. This is really a mistake, especially as some of the people included were over 50, and their medical history was unknown.

Line 171. In reference 6, the definition of “uncertain NCC cases” does not exist. These cases are not NCC and should be included in the results section as “non NCC cases”.

Line 184. PDA? Explain

Line 200: “as at the time of the study, the final diagnosis was not yet known”. I don't understand. Was the scanner not part of the study? This is not at all clear.

Line 207. 249 out of how many people evaluated?

Line 208. The neurologist never saw the patient. Although the authors report this problem as a limitation, it is a reason to invalidate the interest of this study.

Lines 223-226. The results of Ag and Ab detection were part of the diagnostic criteria. These sentences are therefore of little interest.

Lines 230-231. Did the CT scan detect any other brain lesions other than NCC? Please give details. Has surrounding edema or contrast enhancement been detected around any calcified cases?

Table 1. Only one patient with PWSH has been definitively diagnosed with NCC. This should be discussed.

How do the authors explain that none of the three individuals with hydrocephalus suffered from headaches? This is really strange and raises doubts about the proper assessment of “PWSH”. What therapeutic measures were taken for these patients?

Table 2 - Not relevant, as these people do not meet the NCC diagnostic criteria.

PLOS authors have the option to publish the peer review history of their article (what does this mean? ). If published, this will include your full peer review and any attached files.

**Do you want your identity to be public for this peer review?** For information about this choice, including consent withdrawal, please see our Privacy Policy .

Reviewer #1: **Yes:** Ashish Bhalla

Reviewer #2: No

Reviewer #3: No

**Figure resubmission:**

**Reproducibility:**



---

## [Decision Letter · Decision Letter 1]

7 Dec 2025

Prevalence of neurocysticercosis and its characteristics among people with epileptic seizures and progressively worsening severe headaches in 60 villages in three provinces of Burkina Faso

Dear Dr. Carabin,

Thank you for submitting your manuscript to PLOS Neglected Tropical Diseases. After careful consideration, we feel that it has merit but does not fully meet PLOS Neglected Tropical Diseases's publication criteria as it currently stands. Therefore, we invite you to submit a revised version of the manuscript that addresses the points raised during the review process.

Please submit your revised manuscript within by Jan 06 2026 11:59PM. If you will need more time than this to complete your revisions, please reply to this message or contact the journal office at plosntds@plos.org. Please include the following items when submitting your revised manuscript:

We look forward to receiving your revised manuscript.

Kind regards,

Aysegul Taylan Ozkan, M.D., Ph.D.,

Academic Editor

Eva Clark

Section Editor

Shaden Kamhawi

co-Editor-in-Chief

Paul Brindley

co-Editor-in-Chief

**Journal Requirements:**

**Reviewers' Comments:**

Reviewer's Responses to Questions

**Key Review Criteria Required for Acceptance?**

**Methods**

-Are the objectives of the study clearly articulated with a clear testable hypothesis stated?

-Is the study design appropriate to address the stated objectives?

-Is the population clearly described and appropriate for the hypothesis being tested?

-Is the sample size sufficient to ensure adequate power to address the hypothesis being tested?

-Were correct statistical analysis used to support conclusions?

-Are there concerns about ethical or regulatory requirements being met?

Reviewer #2: (No Response)

Reviewer #3: See my comments below

Reviewer #4: Overall, the study shows strong methodological rigor and adequately fulfills the primary requirements for acceptance:

Objectives and Hypothesis

The study’s objectives are clearly defined, accompanied by a precise and testable hypothesis. The justification provided underscores the importance and relevance of the research topic.

Study Design

The selected study design is suitable for addressing the proposed objectives. The methodological structure is well-organized, enabling consistent and meaningful data collection.

Study Population

The characteristics of the study population are clearly outlined and appropriate for evaluating the proposed hypothesis.

Sample Size

The sample size is adequate to ensure sufficient statistical power, which enhances the reliability and credibility of the results.

Statistical Analyses

The statistical approaches employed are suitable and properly executed, consistently reinforcing the study’s conclusions. Data processing is clear, transparent, and methodologically sound.

Ethical and Regulatory Considerations

There are no identified issues related to ethical or regulatory adherence. The research complies with established ethical guidelines and good research practices.

**Conclusion**

In conclusion, the study demonstrates solid methodological quality, appropriate analytical procedures, and full ethical compliance, placing it in a strong position for acceptance.

**Results**

-Does the analysis presented match the analysis plan?

-Are the results clearly and completely presented?

-Are the figures (Tables, Images) of sufficient quality for clarity?

Reviewer #2: (No Response)

Reviewer #3: See my comments below

Reviewer #4: The analysis conducted aligns well with the initial analytical plan, reflecting methodological consistency across the study. The results are presented clearly and thoroughly, allowing readers to readily grasp both the outcomes and their relevance. In addition, the visual materials—such as tables—contribute to the clarity of the data display and reinforcing the interpretation of the findings.

**Conclusions**

-Are the conclusions supported by the data presented?

-Are the limitations of analysis clearly described?

-Do the authors discuss how these data can be helpful to advance our understanding of the topic under study?

-Is public health relevance addressed?

Reviewer #2: (No Response)

Reviewer #3: See my comments below

Reviewer #4: The conclusions are firmly grounded in the data, reflecting the robustness and coherence of the study’s findings. The authors clearly recognize the limitations of their analysis, demonstrating transparency and reinforcing the study’s overall credibility. They also provide a thoughtful discussion on how the results contribute to expanding knowledge in the field, emphasizing the study’s importance within the broader scientific landscape. Additionally, the implications for public health are clearly articulated, highlighting the value of the research in guiding practice and shaping future actions.

**Editorial and Data Presentation Modifications?**

Reviewer #2: (No Response)

Reviewer #3: See my comments below

Reviewer #4: Considering that this work has already been reviewed by three reviewers, many of my doubts and questions have been addressed. The modifications made by the authors in relation to the proposed suggestions and comments clarified my doubts and improved the text. For this reason, my opinion is to accept the article for publication.

**Summary and General Comments**

Reviewer #2: Almost all the comments are appropriately addressed in the manuscript, thanks, no further comments.

Reviewer #3: Although the authors have made efforts to address my concerns, several important issues remain unresolved.

1. I still do not fully understand the rationale behind the “uncertain” diagnostic category. The authors state: “Individuals with possible NCC lesions on neuroimaging which did not meet radiological criteria for definitive or probable NCC according to the diagnostic criteria (4) were categorized as uncertain NCC.” Diagnostic criteria for NCC have been carefully developed by multiple experts, and introducing a new category that is not part of these internationally accepted criteria is not appropriate. What does this new category contribute to the results, and what is its relevance? The statement in the abstract (lines 45–46) is inaccurate: this category is not in accordance with the “internationally recognized diagnostic criteria for NCC.” This category must be deleted.

2. The authors claim that performing contrast CT scans on asymptomatic individuals is not ethical, but this appears to have been done. In Table 1, among the individuals classified as definitive + probable NCC, two subjects reported as symptomatic at first evaluation were later considered asymptomatic by a neurologist. The same applies to one participant categorized as “uncertain” NCC, and it is unclear how many individuals with normal CT scans fall into this situation. Moreover, the added value of contrast CT is never discussed. The authors do not explain how contrast-enhanced CT contributed beyond non-contrast CT, which is particularly relevant given the discussion regarding the relationship between epilepsy, calcifications, and inflammation.

3. The definition used in this study (“PWSH were defined as symptoms being progressively worsening in severity…”) is not comparable to the definition used in reference #14, which evaluated severe headaches caused by migraine or tension-type headache, not progressively worsening symptoms. The same applies to the Zambian study. In the discussion, the authors still claim that PWSH is one of the two most common neurological manifestations.

4. The authors performed cCT scans in 116 PWSH, 109 with epilepsy, and 15 with both conditions. They exclude the “both” group from analysis but include these individuals in each symptom category. Therefore, each category should include an additional 15 participants, resulting in 131 PWSH and 124 PWE. Why then is the reported number of PWE 126 in the manuscript?

Reviewer #4: (No Response)

PLOS authors have the option to publish the peer review history of their article (what does this mean? ). If published, this will include your full peer review and any attached files.

**Do you want your identity to be public for this peer review?** For information about this choice, including consent withdrawal, please see our Privacy Policy .

Reviewer #2: No

Reviewer #3: No

Reviewer #4: No

**Figure resubmission:**
---

## [Decision Letter · Decision Letter 2]

9 Feb 2026

Response to Reviewers
Revised Manuscript with Track Changes
Manuscript

Shaden Kamhawi

co-Editor-in-Chief

Paul Brindley

co-Editor-in-Chief

**Reviewers' comments:**

**Key Review Criteria Required for Acceptance?**

**Methods**

-Are the objectives of the study clearly articulated with a clear testable hypothesis stated?

-Is the study design appropriate to address the stated objectives?

-Is the population clearly described and appropriate for the hypothesis being tested?

-Is the sample size sufficient to ensure adequate power to address the hypothesis being tested?

-Were correct statistical analysis used to support conclusions?

-Are there concerns about ethical or regulatory requirements being met?

Reviewer #3: (No Response)

**Results**

-Does the analysis presented match the analysis plan?

-Are the results clearly and completely presented?

-Are the figures (Tables, Images) of sufficient quality for clarity?

Reviewer #3: (No Response)

**Conclusions**

-Are the conclusions supported by the data presented?

-Are the limitations of analysis clearly described?

-Do the authors discuss how these data can be helpful to advance our understanding of the topic under study?

-Is public health relevance addressed?

Reviewer #3: (No Response)

**Editorial and Data Presentation Modifications?**

Reviewer #3: (No Response)

**Summary and General Comments**

Reviewer #3: Dear authors, just two minor comments:

Please clarify in the first sentence on page 18 (“In our study, this prevalence was estimated at a lower 7.6% (10/131, 95%CI: 3.7%-13.6%)” that your study only looked at one type of headaches (PWSH).

In the first sentences of the last paragraph on page 18 (“In our study, all but one PWSH had calcified NCC. This contrasts with a study conducted in Ecuador where the prevalence of calcified NCC in participants with headaches was estimated at 26.3% (15/57) (14). However, a more recent community-based survey showed larger frequency of lifetime headaches prevalence (4.18, 95% CI: 1.79–9.75) among calcified NCC cases compared to controls without headaches (39)”. Please note that your definition of headache was not the same as that used in the two references mentioned.

PLOS authors have the option to publish the peer review history of their article (what does this mean? ). If published, this will include your full peer review and any attached files.

**Do you want your identity to be public for this peer review?** For information about this choice, including consent withdrawal, please see our Privacy Policy .

Reviewer #3: No

**Figure resubmission:**

**Reproducibility:** To enhance the reproducibility of your results, we recommend that authors of applicable studies deposit laboratory protocols in protocols.io, where a protocol can be assigned its own identifier (DOI) such that it can be cited independently in the future. Additionally, PLOS ONE offers an option to publish peer-reviewed clinical study protocols. Read more information on sharing protocols at https://plos.org/protocols?utm_medium=editorial-email&utm_source=authorletters&utm_campaign=protocols

---

## [Editor Report · Decision Letter 3]

19 Feb 2026

Dear Dr. Carabin,

We are pleased to inform you that your manuscript 'Prevalence of neurocysticercosis and its characteristics among people with epileptic seizures and progressively worsening severe headaches in 60 villages in three provinces of Burkina Faso' has been provisionally accepted for publication in PLOS Neglected Tropical Diseases.

Best regards,

Aysegul Taylan Ozkan, M.D., Ph.D.,

Academic Editor

Eva Clark

Section Editor

Shaden Kamhawi

co-Editor-in-Chief

Paul Brindley

co-Editor-in-Chief

---

## [Editor Report · Acceptance letter]

Dear Dr. Carabin,

We are delighted to inform you that your manuscript, "Prevalence of neurocysticercosis and its characteristics among people with epileptic seizures and progressively worsening severe headaches in 60 villages in three provinces of Burkina Faso," has been formally accepted for publication in PLOS Neglected Tropical Diseases.

Best regards,

Shaden Kamhawi

co-Editor-in-Chief

Paul Brindley

co-Editor-in-Chief
